# Batch Fabrication of a Polydimethylsiloxane Based Stretchable Capacitive Strain Gauge Sensor for Orthopedics

**DOI:** 10.3390/polym14122326

**Published:** 2022-06-08

**Authors:** Karthika Sheeja Prakash, Hermann Otto Mayr, Prachi Agrawal, Priyank Agarwal, Michael Seidenstuecker, Nikolaus Rosenstiel, Peter Woias, Laura Maria Comella

**Affiliations:** 1Department of Microsystems Engineering—IMTEK, University of Freiburg, 79110 Freiburg im Breisgau, Germany; prachi2204@gmail.com (P.A.); priyank.agarwal96@gmail.com (P.A.); woias@imtek.uni-freiburg.de (P.W.); 2Department of Orthopedics and Trauma Surgery, The University Medical Center Freiburg, 79106 Freiburg im Breisgau, Germany; hermann.o.mayr@gmail.com (H.O.M.); nikolaus.rosenstiel@me.com (N.R.); 3Gewebeersatz, Regeneration & Neogenese (G.E.R.N.), Department of Orthopedics and Trauma Surgery, The University Medical Centre Freiburg, 79108 Freiburg im Breisgau, Germany; michael.seidenstuecker@uniklinik-freiburg.de; 4Department of Traumatology and Orthopedics, Pediatric- and Sports-Traumatology, St. Josefskrankenhaus, Artemed Kliniken, 79104 Freiburg im Breisgau, Germany

**Keywords:** batch fabrication, electronic contacting, flexible sensor mold fabrication, PDMS, Carbon black-PDMS, ACL rupture, knee joint

## Abstract

Polymer-based capacitive strain gauges are a novel and promising concept for measuring large displacements and strains in various applications. These novel sensors allow for high strain, well above the maximum values achieved with state-of-the-art strain gauges (Typ. 1%). In recent years, a lot of interest in this technology has existed in orthopedics, where the sensors have been used to measure knee laxity caused by a tear of the anterior cruciate ligament (ACL), and for other ligament injuries. The validation of this technology in the field has a very low level of maturity, as no fast, reproducible, and reliable manufacturing process which allows mass production of sensors with low cost exists. For this reason, in this paper, a new approach for the fabrication of polymer-based capacitive strain gauges is proposed, using polydimethylsiloxane (PDMS) as base material. It allows (1) the fast manufacturing of sensor batches with reproducible geometry, (2) includes a fabrication step for embedding rigid electrical contacts on the sensors, and (3) is designed to produce sensor batches in which the size, the number, and the position of the sensors can be adapted to the patient’s anatomy. In the paper, the process repeatability and the robustness of the design are successfully proven. After 1000 large-strain elongation cycles, in the form of accelerated testing caused much higher strains than in the above-mentioned clinical scenario, the sensor’s electrical contacts remained in place and the functionalities were unaltered. Moreover, the prototype of a patient customizable patch, embedding multiple sensors, was produced.

## 1. Introduction

In recent years, interest has increased in stretchable sensing systems in the field of orthopedics, where the use of stretchable capacitive strain gauge sensors has been investigated for the monitoring and diagnosis of knee laxity caused by the anterior cruciate ligament [1,2]. The anterior cruciate ligament (ACL) tear is one of the most common knee injuries [3]. It destabilizes the knee joint enormously, resulting in anteromedial rotational laxity [4]. 

A promising new approach for the measurement and diagnosis of ACL tear consists of using strain gauges. Strain/motion sensors exist based on several measurement mechanisms [5]: resistive, capacitive, and piezoelectric. The piezoelectric sensors [6] have high sensitivity and stretchability. However, to obtain good performance and functionalities, complex and expensive fabrication methods have to be used. Resistive sensors exhibit high sensitivity, but they show large hysteresis and non-linear electromechanical responses, whereas the capacitive type exhibits small hysteresis, and high repeatability and stretchability [7,8,9,10]. The capacitive approach has been successfully proven on the research level, and a sensor was characterized for the ACL in [4]: The measurements obtained with the methods currently used in the clinic are not repeatable for the diagnosis of knee laxity [11,12,13] and are strongly influenced by the experience of the physician [14,15,16,17]. In [4], the sensor applied over the knee translates the length changes that the skin exhibits during a bone-to-bone displacement into a capacitance variation. However, full validation of highly stretchable smart sensor systems for ACL and general health monitoring has not been possible yet. It would require a reliable batch manufacturing process that allows the mass production of sensors at a low cost. The fabrication process plays a crucial role in making the sensor measurements repeatable and reproducible. That will be the main focus of this paper. Several fabrication methods for flexible capacitive strain gauge sensors in the literature involve cleanroom technologies [18,19,20], such as etching, photolithography, electroplating, and plasma treatment. These complex manufacturing processes cause low yields and high costs. In [21], flexible capacitive strain sensors with different microstructures were developed. The different strain microstructures exhibited issues related to stretchability, a complicated manufacturing process, low yields, and high costs. Alternative low cost solutions are (1) doctor blading [22] into pre-structured molds [23,24], (2) injection or cast molding, and (3) spray coating [25]. In the first technique, doctor blades get liquid PDMS into the cavities of a planar, half-open negative mold, to achieve a structured PDMS device after curing and mold release. This is a fast and low-cost solution. However, it is not reproducible, as the outcome is strongly dependent on the manufacturing operator. Hence, its use in mass production would result in a poor yield. Injection molding is a more repeatable and low-cost fabrication method that is generally used for the structuring of PDMS in microfluidic applications [26,27]. The technique uses a closed mold where liquid PDMS is injected in, to receive a microstructured PDMS device after curing and mold release. However, for PDMS with higher viscosity, as used, e.g., in this study for conductive PDMS, the method is prone to generate molding defects due to incomplete mold filling [28]. The spray coating method [25] would have been unsuitable because of the viscosity of the material used in this study. The drop-casting method followed by the solidification described in [29] to shape a mixture of PDMS and carbon particles was unsuitable, as the uniform surface which is required for this application is not achievable. To overcome the limitations of the low-cost solutions, we propose in this work an innovative, reliable, and simple batch manufacturing process. 

Another challenge for the design of the fabrication process is to develop a solution for embedding a cost-effective, mechanically stable, and rigid electrical contact between highly stretchable elements of a strain gauge sensor and the—naturally rigid—sensor electronics. This is a non-trivial task, as the rigid contact creates a mechanical discontinuity within the stretchable silicone, resulting in a weak spot that is easily breakable. In [30], a multistep process using sputtering and photolithography is proposed for this contact. This method is, however, complex and expensive. Moreover, the adhesion between metal and PDMS is poor. Improvement is possible by fabricating an additional adhesion layer, which, however, requires additional fabrication steps (lift-off and etching) [31]. An alternative method is copper plating on PDMS. However, it causes process-induced cracks due to the strain applied to the copper layer, if the strain on the whole structure is larger than 10% [32].

Further, in medical applications, a strain gauge sensor may have to be designed and fabricated which is patient-specific, i.e., adapted to the anatomy of the patient. This requirement, which is also addressed here, calls for a rapid, design-flexible, and nevertheless cost-efficient fabrication technique for individually customized sensors. A perfect solution to all mentioned issues would allow embedding the sensor into a wearable measurement system, intended as a complex joint platform constituted by sensors and electronics for signal acquisition and data processing. It is an obligatory step to validate any such sensor in the clinic. 

To achieve this goal, in this paper we describe: (1) an innovative and reliable batch fabrication method that not only minimizes the sensor’s mechanical tolerances but also allows for the rapid and patient-specific fabrication of sensing patches; (2) the development of a method for embedding a rigid electrical contact on the flexible sensor that permits a robust connection between the highly flexible sensor elements and the rigid signal processing electronics.

## 2. Functioning Principle of the Sensor

As shown in Figure 1, the sensor consists of two layers: a stretchable substrate, and on top of it, a stretchable conductive layer structured as a capacitive strain gauge [4]. Rigid electrical contacts are embedded into both layers. They are used to connect the sensor with the electronics. The sensor design is based on a comb-shaped capacitive strain gauge, and the capacitance is calculated based on the formula [33] below: $$C=\frac{ƐA}{d}=\frac{{Ɛ}_{0}{Ɛ}_{r}l_{0}t}{d}+C_{F}$$
where Ɛ_0_ is the relative permittivity of vacuum; Ɛ*_r_* is the relative permittivity of dielectric material; *A* is the effective electrode area of the sensor, i.e., the sidewalls of the electrodes facing each other, defined from *l*_0_*t* as the corresponding length and thickness of the comb fingers; *d* is the distance between the fingers and *C_F_* is the sum of the fringe and parasitic capacitance. When the sensor is stretched, the distance between the fingers increases, decreasing a significant part of the capacitance, as given in the equation above.

## 3. Materials and Fabrication Methods

PDMS, obtained from the blend of two base materials, was used for the substrate. Two silicones, Neukasil^®^ RTV-23 and RTV-17 (both from Altropol Kunststoff GmbH, Stockelsdorf, Germany), were mixed in a weight ratio of 10:4 and degassed to release all the air bubbles. For the conductive layer, an electrically conductive PDMS blend, called here C-PDMS (Carbon black-PDMS), was obtained by further mixing this silicone blend with 10.6 wt% [23] carbon black powder ENSACO^®^ 250 P (TIMCAL Ltd., Bodio, Switzerland) [34,35]. The carbon black and PDMS were stirred with a blade mixer at 1200 rpm for 10 min. The obtained mixture was highly viscous, and the texture did not help with the uniform spreading of the material in the mold. Therefore, the viscosity of this mixture was reduced by adding 2.04 g of n-heptane [36].

An alternative silicone material (SYLGARD™184, Dow Corning Corp., Midland, MI, USA) was tested to produce the sensor. However, from the experimental results, the elasticity of the Neukasil silicone turned out to be superior; therefore, that material was preferred.

Two fabrication processes were developed and tested which differ for the sensor contacting method used. In both cases, a rigid contact element was embedded into the C-PDMS, through which mechanical mounting allowed a direct mechanical bond and an electrical contact between the elastic sensor and the rigid printed circuit board (PCB) carrying the sensor electronics. However, in batch manufacturing process I, a screw was used for this connection, whereas in batch manufacturing process II, a mushroom-shaped aluminum pin was embedded for electrical contact. A detailed description and a comparative study of the two processes follow to identify the more suitable one for our purposes. 

### 3.1. Batch Manufacturing Process I

In general, the newly developed batch fabrication method is a multi-step manufacturing process, as shown in Figure 2.

Two molds are necessary for the process. Mold 1 in Figure 2 is used for the manufacturing of C-PDMS sheets. It has a rectangular cavity of 95 × 70 × 0.5 mm milled into an aluminum block. The bottom surface of this cavity is covered with polylactide tape to homogenize the roughness resulting from the milling process. At all edges of the cavity, grooves are milled as sinks for excess PDMS to be collected during the molding process. The mold is closed during the process with a lid made from aluminum, also covered with a thin polylactide tape on the surface. 

In step (1) of the manufacturing process shown in Figure 2, the degassed C-PDMS mixture is poured into the prepared mold to obtain a conductive C-PDMS sheet with a thickness of 500 µm. The quantity of the mixture that is placed into the mold plays a significant role in achieving a uniform height of the sample. Therefore, the C-PDMS mixture is correctly weighed to 4 g per mold and then homogeneously distributed on the cavity surface.

In step (2) of Figure 2, the lid is closed tightly with screws so that the material spreads evenly in the mold due to the pressure applied. Excess C-PDMS is collected in the sinks present around the mold cavity. Then, the C-PDMS layer is cured at room temperature for 24 h. Finally, the C-PDMS sheet is removed from mold 1 and cut into 5 strips. In step (3) of Figure 2, mold 2 is used to assemble 5 C-PDMS strips onto a common substrate foil of electrically insulating PDMS. This mold carries 5 cavities to place the cut-out carbon black strips. The cavities for the C-PDMS of dimension 59.8 × 11.5 × 0.5 mm are milled into the bottom of a larger cavity into which pure PDMS is poured after mounting the C-PDMS strips. The material, evenly pressed by the lid, is cured at room temperature for 8 h, resulting in a cross-linking of the conductive and non-conductive layers of PDMS. This contributes to good adhesion between the PDMS and C-PDMS.

The C-PDMS strip sheets with PDMS as an insulator layer are removed from mold 2, giving the resulting sheet in Figure 3. The sensor strips are then cut to shape with a doctor blade.

The C-PDMS strip is then structured into an interdigital comb shape through lasering, using a Nd:YAG laser with a wavelength of 1064 nm (DPL Smart Marker II, ACI Laser GmbH, Nohra, Germany) [4]. The final sensor structure consists of 279 fingers and two enlarged contact areas, as shown in Figure 4. Additionally, the connecting sidelines for each set of fingers are enlarged by a width of 2 mm to provide for lower electrical resistance. The total length and width of one strain gauge sensor are 65 and 11.5 mm, respectively. The strain-sensitive area covered by the capacitive sensor is 55.8 mm long and 7.5 mm wide.

To embed the electrical contacts, the contact pad area of the sensor is pierced, and a screw is inserted from the bottom of the sensor. Metallic top and bottom washers are used to tighten the sensor to the contact terminals of the PCB with a bolt, as shown in Figure 5a. The screw reaching through the metal plates applies a normal force to the contact point with the C-PDMS. This restricts the movement of the C-PDMS at one end along the displacement direction of the sensor. In this way, a robust mechanical and electrical connection is guaranteed between the sensor and PCB (see Figure 5b). The head of the screw is then covered with Kapton foil to isolate the contact pads. 

### 3.2. Batch Manufacturing Process II

For this manufacturing process, the molds with holes are used. They differ from the previous one by the 10 boreholes of 1.05 mm diameter and 1.40 mm depth that are used to hold rigid electrical contacts during the molding of a C-PDMS layer. The rigid electronic contacts are produced from aluminum with a CNC machine and were designed as shown in Figure 6. They exhibit a mushroom-like shape with small holes in their crown to ensure good mechanical fixation and electrical connection when embedded into the C-PDMS layer. Moreover, to guarantee their stable connection with the C-PDMS during the sensor elongation as well, the contact pads are coated with a primer material (NuSil SP-120, Silicone Primer, Songhan Plastic Technology Co., Ltd., Shanghai, China).

Figure 7 shows manufacturing process II in detail, which corresponds almost entirely with manufacturing process I. The only difference is that in step (1) after the C-PDMS is poured into mold 1, the primer-coated electronic contacts shown in Figure 6 are pierced into the material where the boreholes are in the mold. While the slender foot of the contact is placed into the boreholes in the mold, the crown will extend into the mold cavity to be surrounded with C-PDMS from all sides and through their holes.

The resulting PDMS sheet is shown below in Figure 8.

Figure 9 shows the sensor resulting from the laser machining of the capacitive sensor structure with 279 fingers.

## 4. Experimental Test

To validate the process and prove its repeatability, three C-PDMS sheets were manufactured using the newly developed batch fabrication method. Their surface roughness and height tolerance were measured using a profilometer, by scanning the C-PDMS sheet along x and y-directions.

To further analyze the repeatability of the production process, two sensors per C-PDMS sheet (Batch 1, Batch 2, Batch 3) were characterized, and the results are compared. Every sensor was subjected to 10 mm stretching and relaxing of the sensor at a constant step size of 1 mm, while the sensor reading was recorded.

The sensor was stretched at 3 mm/sec from one step to the next. The measurements were performed by stretching the sensor with a linear stage (High Precision, LS-110-Micos, Irvine, CA, USA) having 0.2 µm resolution, and data were sampled using a time-to-digital converter built on a microcontroller CC2652R1 (Simple Link™ 32-bit Arm Cortex-M4F multiprotocol 2.4 GHz wireless MCU, Texas Instruments, Freising, Germany). The receiver sensitivity (dBm) of the chip used is 100 dBm for 802.15.4 (2.44 GHz). Hence, the capacitance variation of the sensor was read in terms of the time necessary to discharge the capacitance (clock ticks). The working principle of the chip is the following. The chip charges the unknown capacitance with a constant current until a defined voltage and counts the time of charge in terms of clock ticks (one tick per clock period necessary). The capacitance value can then be defined using the following formula:$$I=C\;\cdot \;\frac{dv}{dt}$$
in which I is the charging current and dv is the voltage until the capacitance is charged. Both values are fixed and given in the datasheet. dt is calculated as clock ticks per clock period.

An additional test was executed to measure the robustness of the electric contacts with the screw (batch manufacturing process I) and the aluminum contact (batch manufacturing process II). This was carried out by subjecting the sensor patch to 10 and 50 mm stretching and releasing at 3 mm/sec for 500 motion cycles and afterward proving its functionality by analyzing the capacitance variation against the frequency with the help of an LCR meter (HM8118—LCR-Meter, Labortisch, 100 MOhm, 100 kH, 0.1 F, 200 kHz, Rohde and Schwarz, Mainhausen, Germany) which has an accuracy of 0.05%. The schematic diagram of the measurement setup is shown below in Figure 10.

## 5. Result and Discussion

### 5.1. Surface Analysis of C-PDMS Sheets

The surface roughness and the height tolerance of three C-PDMS sheets obtained with the newly developed production method were analyzed using a 3D profilometer (Keyence VK-X1000, Osaka, Japan). The average step thickness, obtained by scanning the C-PDMS sheet in the x-direction (horizontal profiling) was 543 µm for Batch 1, 515 µm for Batch 2, and 506 µm for Batch 3. The results of the horizontal profiling are shown in Figure 11a.

The average step thickness obtained by scanning the C-PDMS sheet in the y-direction (vertical profiling) was 500 µm for Batch 1, 524 µm for Batch 2, and 500 µm for Batch 3. The results of the vertical profiling are shown in Figure 11b. The results show tolerance of the C-PDMS thickness. This tolerance is critical for the production method, as the sensor height influences the selection of the laser parameters and the latter capacitance. The procedure to find these parameters is time- and cost-consuming and for this reason not often undertakable. However, the measured tolerance was <60 µm, for which the laser parameters stayed constant for all the sensors of different batches.

### 5.2. Scanning Electron Microscope Images (SEM)

The SEM images were taken for the strain gauges prepared from batch manufacturing processes I and II. Figure 12 below shows the cross-sectional image of the C-PDMS finger electrodes performed with SEM microscope (Scios 2 HiVac, FEI). 

Figure 12a shows the carbon black particle distribution forming a network, which allows a reliable conductive path in the PDMS matrix in batch manufacturing process I. Figure 12b shows the carbon black particle distribution forming a network with a reliable conductive path in the PDMS matrix in batch manufacturing process II. Figure 12c shows the cross-section of C-PDMS from top to bottom to observe the homogeneity of the particle distribution along with the material’s thickness. Figure 12d shows the size of the carbon black particles in the PDMS matrix.

### 5.3. Process Validation through Characterization

Figure 13 shows the characteristic (clock ticks against displacement) of sensor I in Batch 1. The sensor was subjected to five motion cycles.

In the above characterization, perfect correspondence between different cycles of measurement of the sensor was observed with an RMSE of 0.33. 

Figure 14 shows the time-dependent capacitance variation. The sensor was stretched at the speed of 3 mm/sec—hence, 1 mm every 333 ms.

Figure 15 shows the characteristics of the analyzed sensors in Batch 1, Batch 2, and Batch 3.

A small deviation, even if minimal, was observable between the characteristics of the sensors belonging to the same batch. To better analyze this behavior and understand its source, for Figure 16 the data of all the sensors analyzed were concatenated and a second-order polynomial fit was executed. 

Table 1 shows the parameters of the second-order polynomial fit. The R-square value close to 1 determines a good fitting of all the data with the chosen model. In Figure 16, it can be observed that the deviation of every sensor’s characteristic from the fitting curve is almost constant. The results of different RMSE values are shown in Table 2.

The reason for this behavior is that, as the material of the sensor substrate is highly elastic, a clamping of all sensors at the same position on the linear stage is not feasible. Therefore, an unintended stretching of the sensor by fixating causes this deviation between the two sensor curves. As for this work, however, a displacement variation is needed rather than an absolute value. This deviation is an issue that can be solved by subtracting the starting value from all the measurements.

The good fitting of all the data with the same polynomial confirms the reproducibility of the sensors and the reliability of the newly developed manufacturing method.

### 5.4. Analysis of Electrical Contact Robustness and Reliability

The sensor produced with batch manufacturing process I (see Figure 2) withstood the 10 mm stretching without any damage for 500 cycles. However, during the 50 mm stretch and release, the sensor was torn off from the contacting base after 300 cycles. 

The sensor developed with batch manufacturing process II (see Figure 7) withstood the 10 mm stretching for 500 motion cycles. It also withstood the 50 mm stretching for 500 motion cycles, and after 1000 motion cycles, it was still intact, though with a slight crack in the contact pad area. Figure 17 shows the adhesion of the electric contact and the sensor at the end of the test.

With elongations of 10 and 50 mm, these tests are beyond the elongation values expected in knee laxity measurements. Therefore, this test can be regarded as accelerated testing and gives a good prognosis for long lifetimes under lower strain conditions.

Figure 18 shows the capacitance against frequency before and after 500 and 1000 motion cycles in the frequency range from 0.5 to 1.5 kHz.

The measurement results show good correspondence between the two characteristic curves, proving the contact robustness. The use of the primer proved to be the determinant ensuring strong adhesion between the metallic electric contact and the conductive C-PDMS polymer. 

Figure 18 shows the bandwidth of the developed sensor. The bottleneck of the sensor bandwidth is the period of charging and discharging of the sensor. The charging time is obtained by supplying a constant current and measuring the time necessary to reach the maximum voltage. The sensor charging time is 103 µs. The discharging time is in the range of the ns and is obtained by connecting the hot electrode of the capacitance to the ground. As the sensor resistance influences the sensor time response a larger amount of carbon particles is desirable to reduce it. However, a large amount of carbon particles influences the gauge factor negatively. The percentage of carbon black used in this work permits material elasticity suitable for the application and a parasitic resistance which still allows the capacitance charge and discharge with a frequency of 1 kHz.

### 5.5. Fabrication of Patient-Customizable Sensing Patches

Both batch manufacturing processes can be used to rapidly prototype patient-customizable patches for knee laxity measurements, and even embed multiple stretchable capacitive strain gauge sensors. The size, the number, and the position of the sensors are degrees of freedom, which can be adjusted based on the anatomy of the patient. 

Additionally, both manufacturing processes can be executed completely, from step (0) to step (3), for the realization of a complete sensor array that can be applied as a single patch without the need for assembling several different sensors. Figure 19 shows an example patch for the application considered: the measurement of knee laxity. The main purpose is to measure and analyze the influence of ligament movements at multiple positions on the knee, and the sensors’ positioning could be adjusted based on the patient. Data from multiple sensors improves data reliability.

## 6. Conclusions

A novel batch fabrication method was developed which permits the fast and low-cost production of geometrically reproducible, robust, and highly elastic capacitive strain gauge sensors.

The process was validated, and its reliability was successfully proven by measuring the thickness of the C-PDMS sheet. The height tolerance obtained with the newly developed process is <60 µm with an average total height of approximately 500 µm. As all the batches were produced with the same laser parameters, it was proven that the obtained tolerance is in a range that does not invalidate their selection and does not interfere with the repeatability of the production method. Additionally, the capacitance values of sensors obtained from these slightly different C-PDMS layers showed negligible deviations in their characteristics.

To further validate the process’s repeatability, two sensors from three different batches were characterized. The obtained characteristics have the same quadratic behavior and can be successfully fitted with the same polynomial. This means that all the sensors show the same variation to a given elongation.

A difference was observed in the capacitance of the sensor before the stretching. This was further confirmed by the RMSE, which was different for every sensor characteristic. This effect occurred due to the highly elastic behavior of the material and due to the properties of the measurement setup used for characterization. When the sensor is fixed on the linear stage for the measurement, an undesired stretching of the material can happen and will generate a difference among the measured starting values.

This paper presented a reliable solution for connecting the highly flexible and stretchable sensor to the—naturally rigid—electronics for signal processing. It was proven that the two developed solutions, which consist of embedding rigid contacts into the PDMS layer during the sensor manufacturing, provide a robust connection. Between the two solutions, the specifically designed aluminum contact coated with PDMS primer gave more robust adherence. A sensor with this contact that was subjected to 500 and 1000 motion cycles did not show any dysfunctionalities at the end of the test.

No long-term aging test of the sensor was executed in this work. However, for hygienic reasons, it would be desirable to change the sensor often. The eventual variation in stretch performance over time is not relevant for our purposes. Additionally, the exaggerated test conditions employed may be regarded as accelerated testing. They allow us to assume a longer sensor lifetime under more moderate conditions. 

The main novelty of this paper lies in the development of a repeatable fabrication method that allows the batch production of sensors ready for clinical trials. The challenges given by its use in a medical trial were not addressed and will be the object of a follow-up publication.

## Figures and Tables

**Figure 1 polymers-14-02326-f001:**
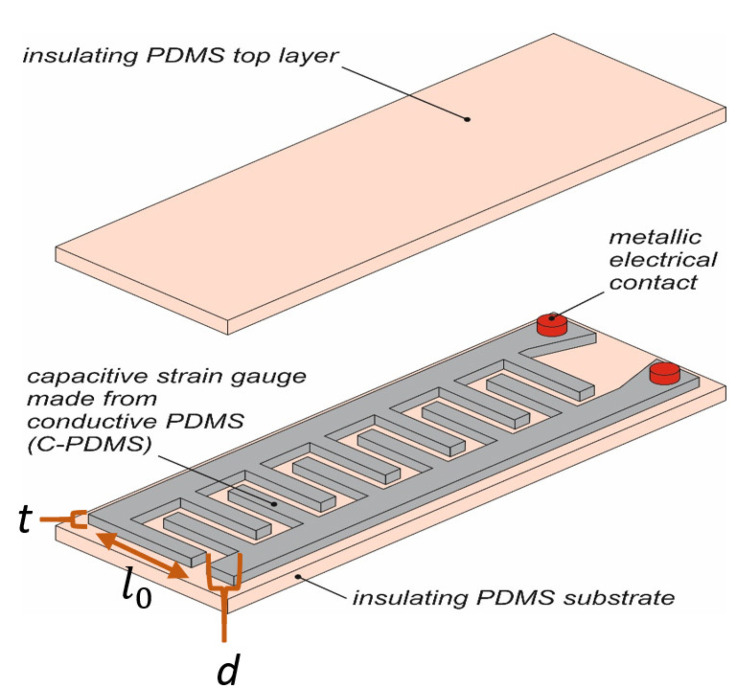
3D schematic of the capacitive strain gauge sensor with embedded rigid contacts. The top layer is lifted for better visibility of the internal structures.

**Figure 2 polymers-14-02326-f002:**
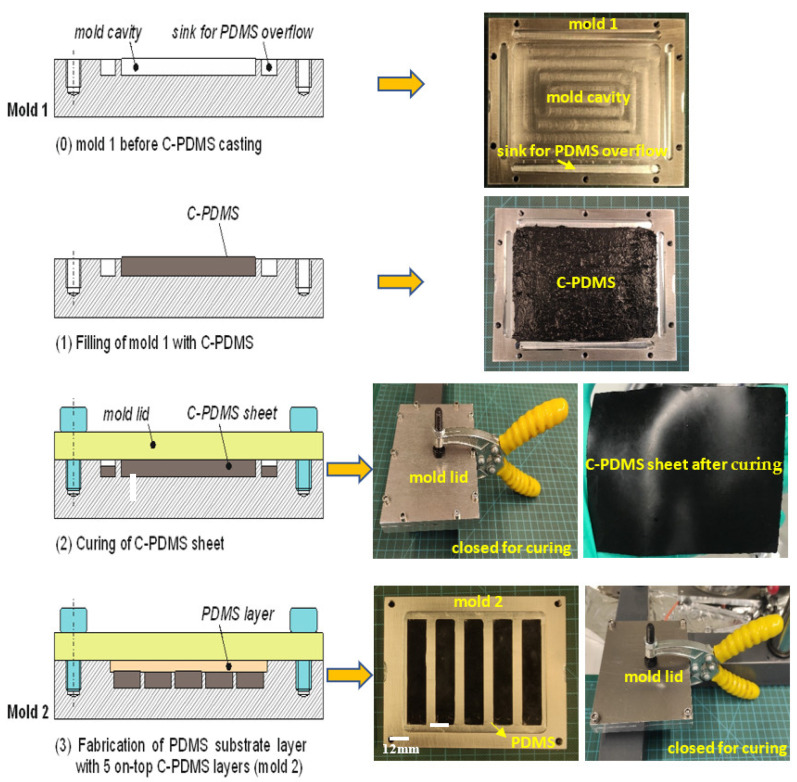
Batch manufacturing process I.

**Figure 3 polymers-14-02326-f003:**
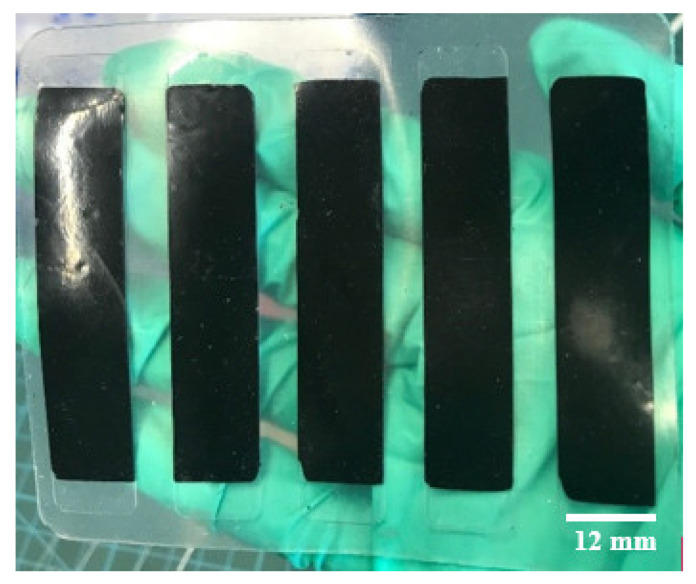
C-PDMS strip sheets embedded into the PDMS layer.

**Figure 4 polymers-14-02326-f004:**
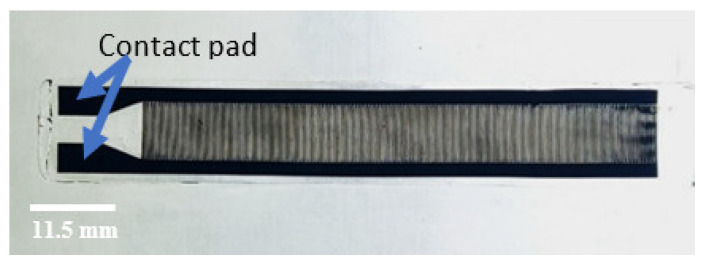
Final sensor structure with 279 lasered fingers.

**Figure 5 polymers-14-02326-f005:**
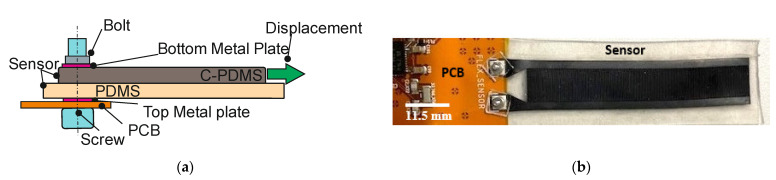
(**a**) Cross-sectional view of the connection between sensor and PCB using a screw. (**b**) Photograph with a top view of the connection between the sensor and the PCB.

**Figure 6 polymers-14-02326-f006:**
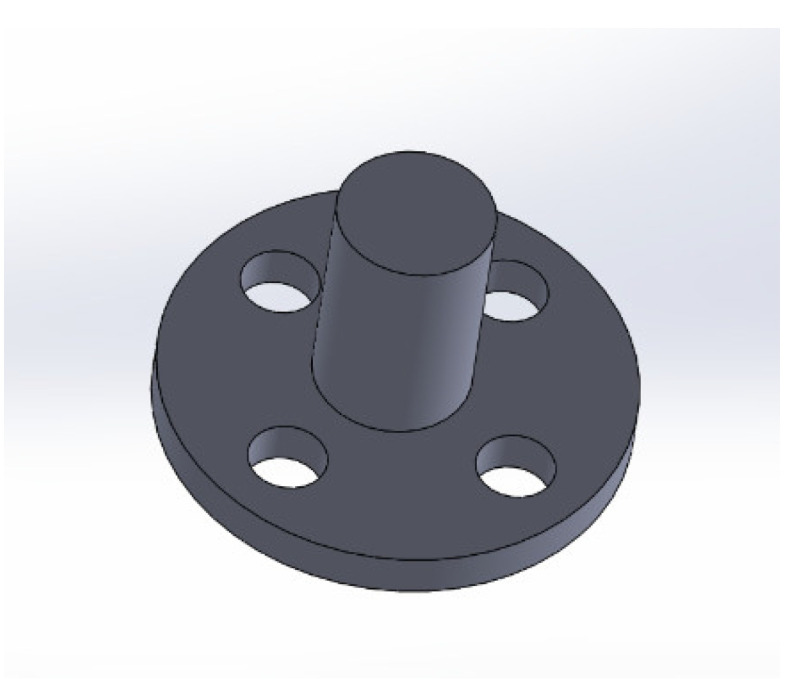
CAD drawing of an aluminum rigid electronic contact designed to be embedded in the flexible strain gauge sensor for contacting the C-PDMS layer. Crown height 0.3 mm, crown diameter 3 mm, holes in the crown 0.5 mm diameter, foot height 1.5 mm, foot diameter 1 mm.

**Figure 7 polymers-14-02326-f007:**
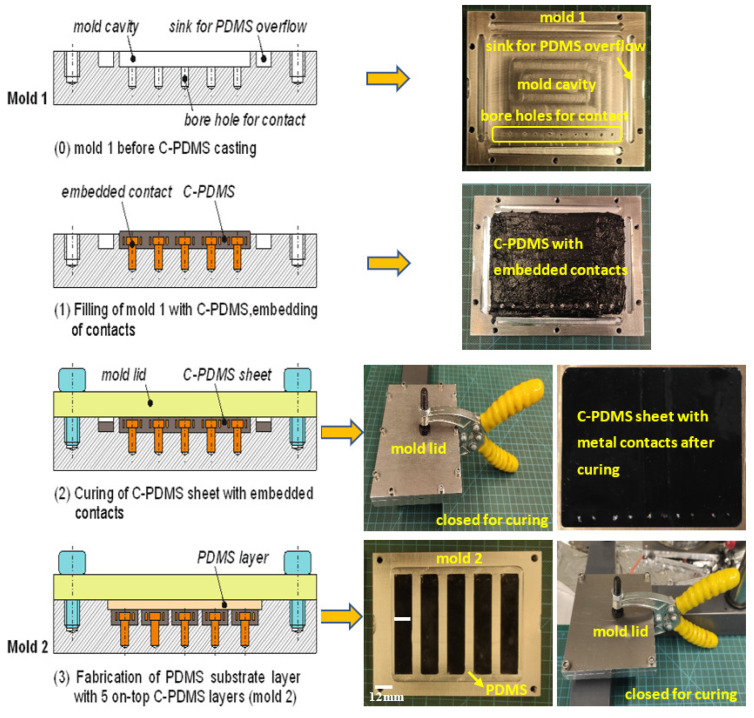
Batch manufacturing process II.

**Figure 8 polymers-14-02326-f008:**
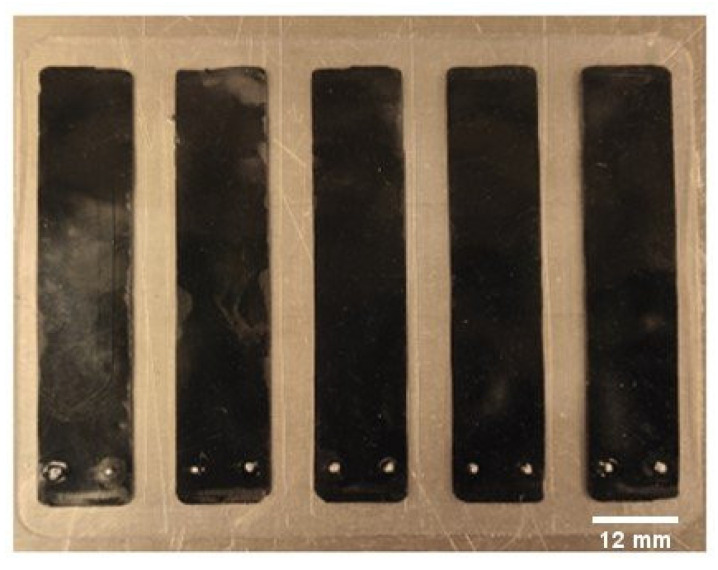
C-PDMS strip sheets are embedded into a PDMS layer with metal contacts.

**Figure 9 polymers-14-02326-f009:**
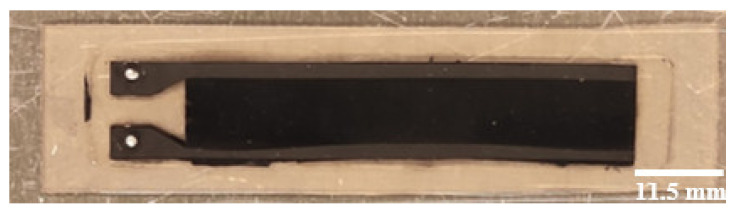
Flexible capacitive strain Gauge sensor produced with batch manufacturing process II.

**Figure 10 polymers-14-02326-f010:**
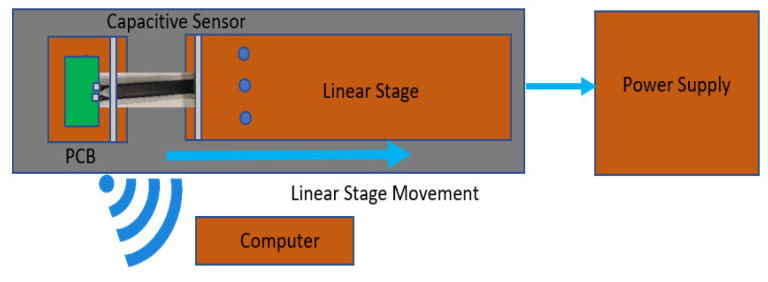
Schematic diagram of the measurement setup.

**Figure 11 polymers-14-02326-f011:**
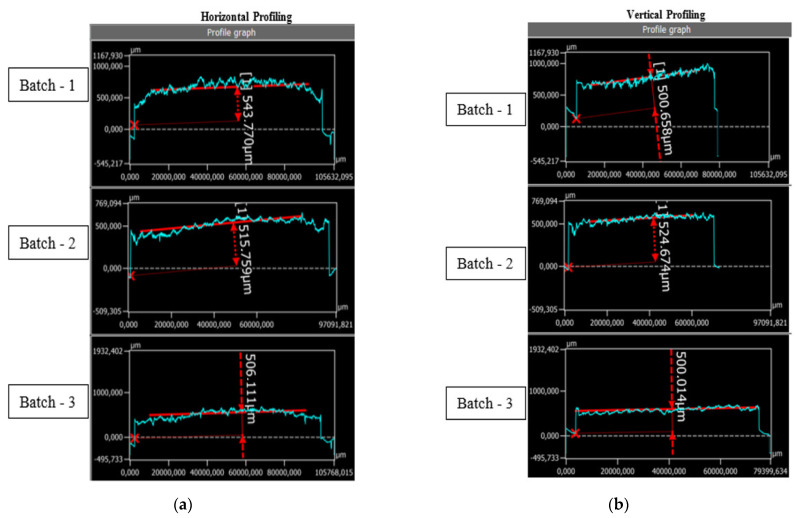
The average step height of a C-PDMS sheet of three different batches as measured with a 3D profilometer (Keyence VK-X1000): (**a**) horizontal profiling and (**b**) vertical profiling.

**Figure 12 polymers-14-02326-f012:**
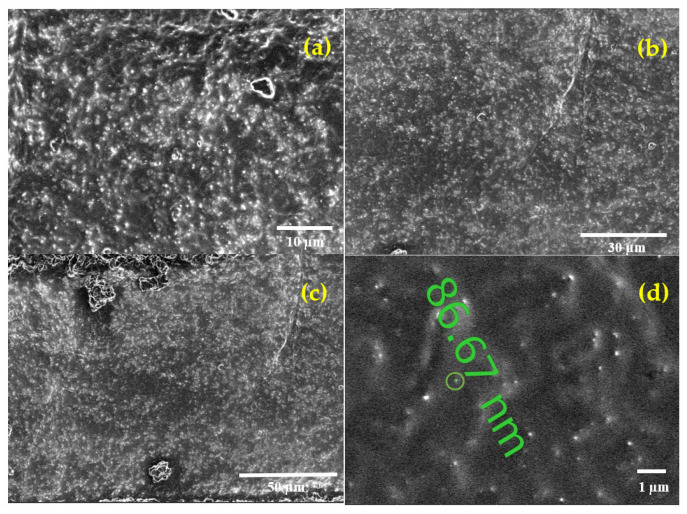
Cross-section of carbon black particle distribution in the PDMS matrix: (**a**) batch manufacturing process I; (**b**) batch manufacturing process II. (**c**) Vertical cross-section image with top and bottom surface of C-PDMS. (**d**) The particle size of carbon black powder in PDMS matrix.

**Figure 13 polymers-14-02326-f013:**
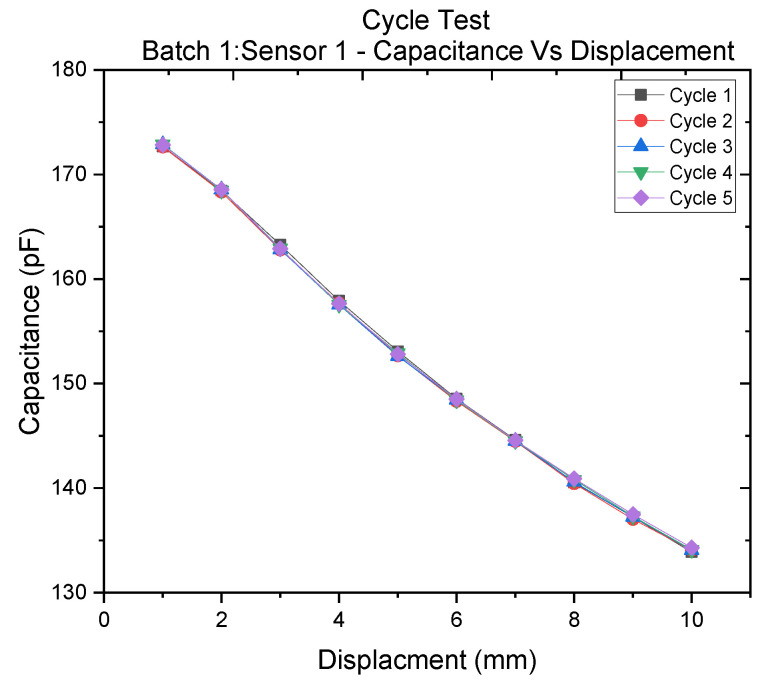
5 Clock ticks against displacement of sensor 1 in batch 1, taken over 5 motion cycles.

**Figure 14 polymers-14-02326-f014:**
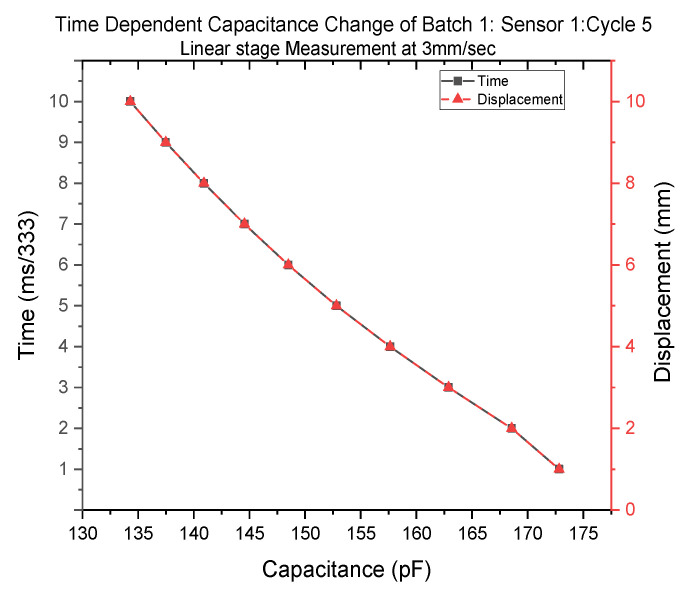
Time-dependent capacitance change of Batch 1: sensor 1: cycle 5.

**Figure 15 polymers-14-02326-f015:**
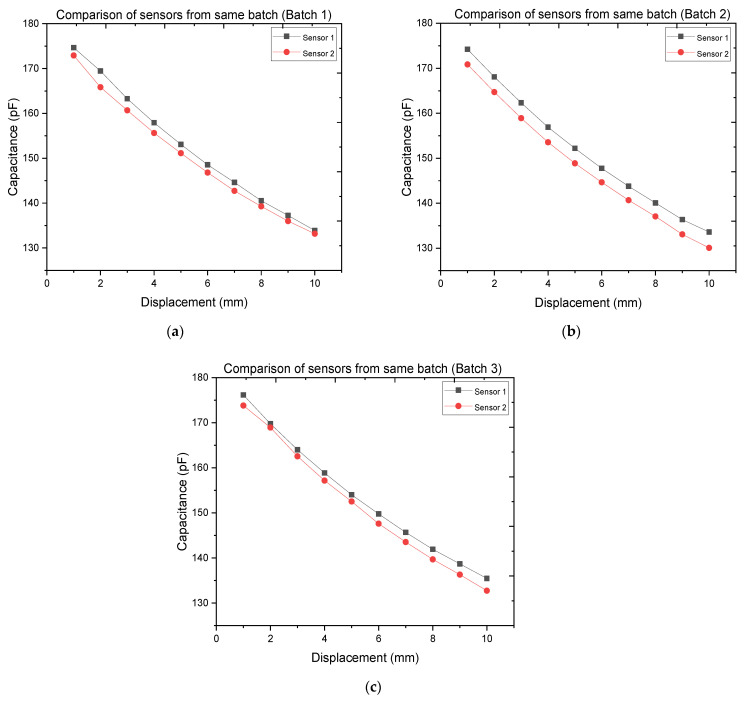
Clock ticks against the displacement of two sensors (sensors 1 and 2) belonging to (**a**) Batch 1, (**b**) Batch 2, and (**c**) Batch 3.

**Figure 16 polymers-14-02326-f016:**
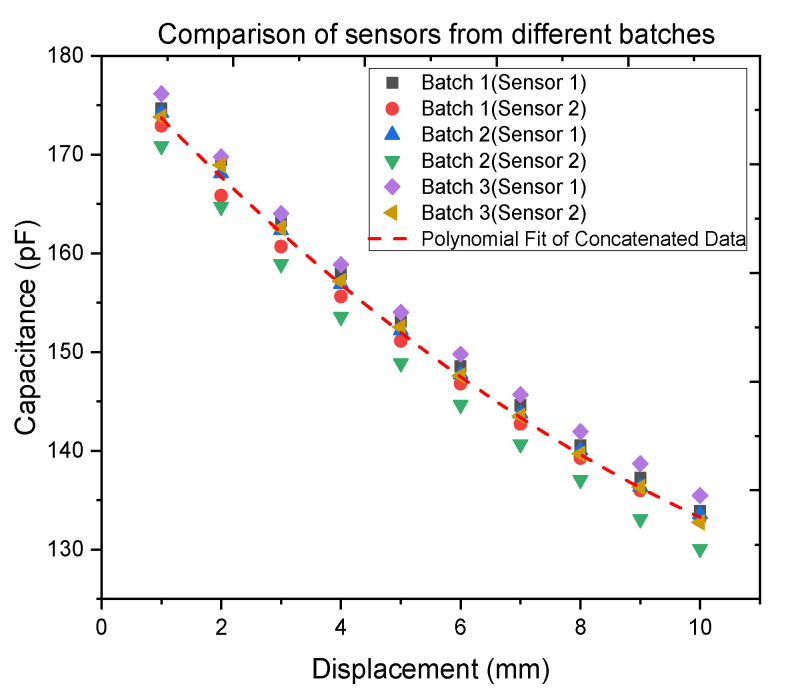
Clock ticks against displacement of two sensors (sensor 1 and 2) belonging to Batches 1, 2, and 3. The data have been concatenated and fitted with a polynomial.

**Figure 17 polymers-14-02326-f017:**
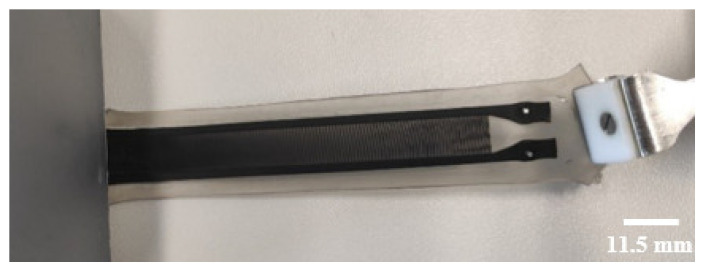
Sensor patch developed with batch manufacturing process II after 1000 motion cycles.

**Figure 18 polymers-14-02326-f018:**
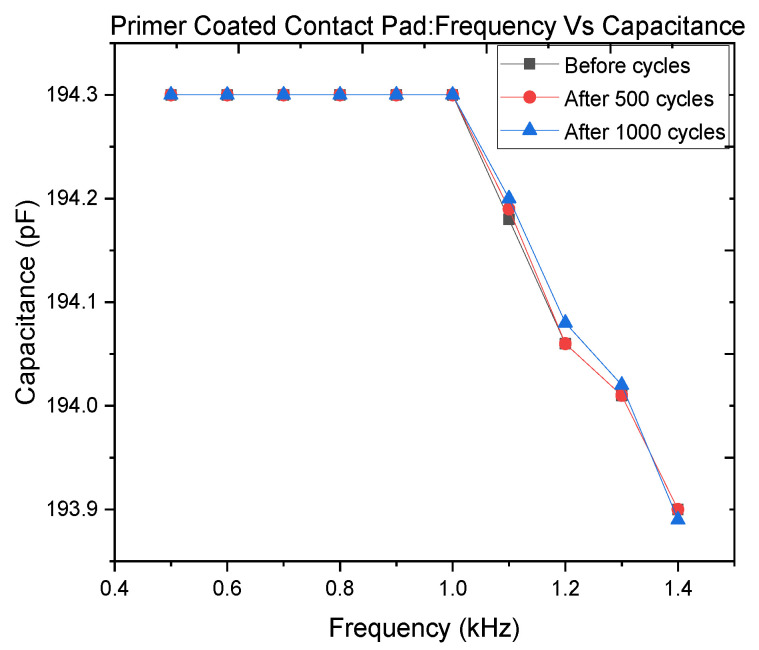
Capacitance against the frequency of the sensor developed with batch manufacturing process II before and after 500 and 1000 motion cycles.

**Figure 19 polymers-14-02326-f019:**
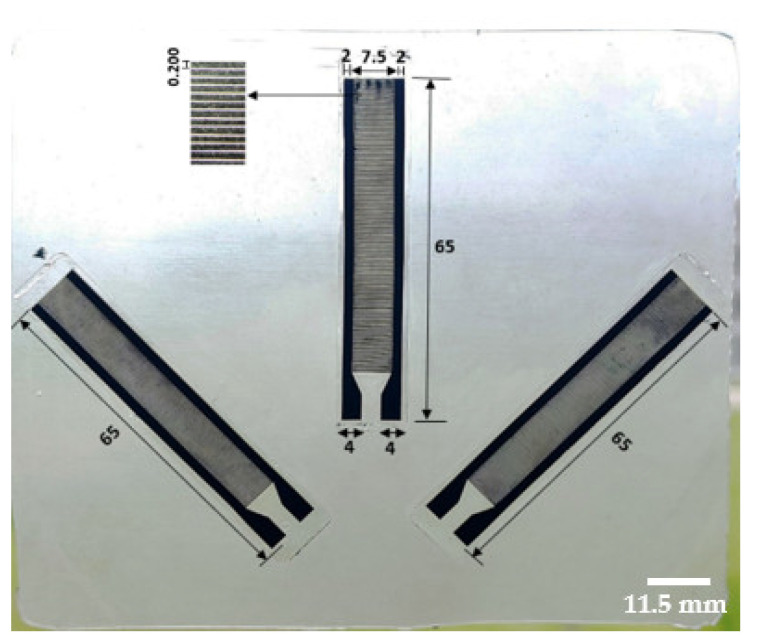
Example of a patient-customizable sensing patch embedding multiple stretchable capacitive strain gauge sensors.

**Table 1 polymers-14-02326-t001:** Second-order polynomial fit parameter values.

Parameters	Values
Equation	$y=\mathrm{Intercept}+B1\;{*X}^{1}$ $+B2\;{*X}^{2}$
Plot	Concatenated Data
Intercept	180.07 ± 0.81
B1	−6.55 ± 0.34
B2	0.19 ± 0.03
Residual sum of squares	164.13
R-Square (COD)	0.98

**Table 2 polymers-14-02326-t002:** Root mean square error (RMSE) of the data measured with sensors 1 and 2 in Batches 1, 2, and 3.

Batches	Sensors	RMSE
Batch 1	Sensor 1	1.12
	Sensor 2	0.96
Batch 2	Sensor 1	0.32
	Sensor 2	2.99
Batch 3	Sensor 1	2.21
	Sensor 2	0.49

## Data Availability

The data presented in this study are available on request from the corresponding author. The data are not publicly available due to further ongoing research and experiments.

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
