# Peer review of "Batch Fabrication of a Polydimethylsiloxane Based Stretchable Capacitive Strain Gauge Sensor for Orthopedics"

_polymers, 2022, doi:10.3390/polym14122326_

Round 1

Reviewer 1 Report

Prakash et a. reported the batch fabrication of PDMS based stretchable capacitive strain gauge sensor. The authors provided a simple process for strain gauge sensor fabrication. Therefore, I recommend that the manuscript publishing after a major revision.

  1. The author used “Orthopedics” in the title. But I did not find related results in the manuscript. Do authors want to say the stretchable capacitive strain gauge sensors have potential applications in Orthopedics?
  2. How about the bonding stability between C-PDMS and PDMS?
  3. Please add scale bars in Figures 2, 3, 5, 7-9, 15, and 17.
  4. Can the authors further provide SEM images of C-PDMS to verify the dispersion of carbon black powder in C-PDMS?
  5. To enrich the background of flexible electronics, the author should add more discussions on smart sensing systems. The following may be useful, such as The Innovation 2022, 2, 100168; Advanced Functional Materials 2020, 30, 2003491; Advanced Materials 2019, 31, 1805921; Advanced Materials 2019, 32, 1901981; Frontiers in Chemistry 2019, 7, 461.

Reviewer 2 Report

The manuscript can only be accepted if the authors answer the following comments.

  1. More background on the sensors is needed. It should include piezoelectric, triboelectric, resistive and capacitive sensor with their pros and cons.
  2. There is no scale bar in the images of Figure 2, 3, 5b, 7, 8, 9, 15, 17,
  3. Why 10.6% of carbon black was mixed with PDMS for fabrication of conductive PDMS. The I-V curve and quantification of conductivity is needed.
  4. Why PDMS was cured at room temperature? In generally it is cured at 60 deg or 80 deg temperatures. Otherwise the material becomes sticky and sometimes it not cross-linked well which affect the materials stretchability. This manual process is not desirable for batch fabrication.
  5. What is the concentration of n-heptane used for decreasing the viscosity of C-PDMS?
  6. Which polymer tape is used for homogenized the roughness?
  7. The time dependent capacitance change data is needed for each displacement and also for 500 and 1000 cycles.
  8. Also, elongation dependent capacitance change data is needed for each displacement.
  9. The SEM or microscopic images are needed for strain gauge sensor prepared in batch fabrication process I and II where the comb shape structure should be clearly observed.
  10. What are the sensitivity of the devices?
  11. The experimental data for the prototype patient-customizable patches for knee laxity measurements is needed.

Round 2

Reviewer 1 Report

I recommend the paper to be accepted for publication. but The English writing is also not satisfactory, some of the sentences are hard to read and follow.

Reviewer 2 Report

The manuscript is ready to accept.